# Adoption and continued use of mobile contact tracing technology: multilevel explanations from a three-wave panel survey and linked data

Laszlo Horvath  ,[1,2] Susan Banducci,[2] Joshua Blamire,[3] Cathrine Degnen,[4] Oliver James,[2] Andrew Jones,[5] Daniel Stevens,[2] Katharine Tyler[6]

[1]Politics, Birkbeck College, University of London, London, UK
[2]Politics, University of Exeter, Exeter, UK
[3]Institute for Community Research and Development, University of Wolverhampton, Wolverhampton, UK
[4]School of Geography, Politics and Sociology, Newcastle University, Newcastle upon Tyne, UK
[5]Faculty of Laws, UCL, London, UK
[6]Sociology, Philosophy and Anthropology, University of Exeter, Exeter, UK

**Correspondence to**
Dr Laszlo Horvath;
L.Horvath@bbk.ac.uk

## ABSTRACT

**Objective** To identify the key individual-level (demographics, attitudes, mobility) and contextual (COVID-19 case numbers, tiers of mobility restrictions, urban districts) determinants of adopting the NHS COVID-19 contact tracing app and continued use overtime.

**Design and setting** A three-wave panel survey conducted in England in July 2020 (background survey), November 2020 (first measure of app adoption) and March 2021 (continued use of app and new adopters) linked with official data.

**Participants** N=2500 adults living in England, representative of England's population in terms of regional distribution, age and gender (2011 census).

**Primary outcome** Repeated measures of self-reported app usage.

**Analytical approach** Multilevel logistic regression linking a range of individual level (from survey) and contextual (from linked data) determinants to app usage.

**Results** We observe initial app uptake at 41%, 95% CI (0.39% to 0.43%), and a 12% drop-out rate by March 2021, 95% CI (0.10% to 0.14%). We also found that 7% of nonusers as of wave 2 became new adopters by wave 3, 95% CI (0.05% to 0.08%). Initial uptake (or failure to use) of the app associated with social norms, privacy concerns and misinformation about third-party data access, with those living in postal districts with restrictions on mobility less likely to use the app. Perceived lack of transparent evidence of effectiveness was associated with drop-out of use. In addition, those who trusted the government were more likely to adopt in wave 3 as new adopters.

**Conclusions** Successful uptake of the contact tracing app should be evaluated within the wider context of the UK Government's response to the crisis. Trust in government is key to adoption of the app in wave 3 while continued use is linked to perceptions of transparent evidence. Providing clear information to address privacy concerns could increase uptake, however, the disparities in continued use among ethnic minority participants needs further investigation.

## INTRODUCTION

As a tool in national COVID-19 Track and Trace systems, mobile contact tracing apps automate the process of contact tracing by

---

**Strengths and limitations of this study**

► Our data captures reported behaviour at two points to assess within-subject changes over time.

► Our results are based on a large, nationally representative sample, as opposed to the convenience and limited-N samples of previous contact tracing studies.

► Integrating demographic/structural and attitudinal explanations relating to technology acceptance with questions adopted from the results of a deliberative poll.

► Studied population is England (see section 2.3) where overall mobility is restricted in wave 3 during national lockdown, allowing for limited opportunities for app usage for example, venue check-ins.

► Drawing on our findings, an ethnic minority booster sample will in the future allow us to better understand inequalities across and within diverse ethnic populations.

---

sending users a notification of possible exposure to the virus, along with health advice. Public acceptance is key to efficiency: Several studies have shown that the app's ability to suppress the epidemic depends on the level of overall uptake. An early estimate indicated app usage of 56%[1] could have helped to avoid a second nationwide lockdown in the UK. Another study indicated 15% uptake would decrease the death toll if combined with effective human contact tracing.[2] On the other hand, the rejection of contact tracing apps by some may suggest that the government failed to secure the public's trust that is crucial for compliance with restrictions on mobility and social contact.[3 4]

Current evidence about who uses contact tracing apps and why is limited in several ways. First, prior to their introduction, studies were only able to measure intention to use apps because they had not been developed and rolled out.[5] The studies relied on

**BMJ**

experimental scenarios looking at the potential properties of the apps that could influence adoption, such as data storage and sponsors.[6 7] Second, of the limited observational evidence available, studies have been restricted to convenience sampling which tends to overestimate adoption rates.[8] A recent study looking at user feedback on Google Play fails to capture nonusers entirely.[9] Important qualitative work has identified key areas of citizen concern (eg, transparency and the needs of vulnerable groups[10] or social norms or pressure[11]) but the distribution of these concerns remains to be investigated at the national level. Third, studies have been limited to exploring adoption at a single time point given the relatively short time since the roll-out of the technology in many countries. Continued use and drop-out rates, thus, remain to be investigated. We include additional notes about our theoretical expectations of the relevant predictors in section 2.3.

In this study, our objective is to address these limitations with a large-scale multiwave study in England, drawn from a probability-based research panel, with representative sample demographics. We measured adoption of contact tracing apps first in November 2020 and again in March 2021. To explain adoption and continued use, we link data from this survey (demographics, attitudes and reported behaviour) with their postal districts' COVID-19 case numbers, urban versus rural majority population, as well as policy restrictions on social gatherings and mobility. Our model specification is informed by the literature on Technology Acceptance particularly of health technology, trust and findings from a deliberative public forum with UK residents.[10] Our predictor on views of the app in respondents' social networks echoes additional fieldwork insights reported recently.[11]

We report uptake at 41% in November 2020 with a 12% drop-out rate by March 2021, and that 7% of nonadopters in November 2020 had installed the app by March 2021. Of the predictors of the uptake, we find that individual-level attitudinal measures best capture the reasons why some adopted the technology while others have not (privacy and norms). We also report concern and misinformation about third party data access among nonusers, and that trust in government was a significant predictor of new adoption in March 2021 which, we speculate, could be related to the severity of the January–March 2021 wave and/or perceptions about the UK Government's early success in its vaccination programme. Explaining continued use specifically, we highlight the role of perceived usefulness and concern about transparent evidence.

## Contact tracing in the UK
The roll-out of the government-backed NHS COVID-19 app on 24 September 2020, by National Health Services (NHS) England and Wales, makes the UK a relatively late adopter of digital contact tracing, 6 months behind their first recorded use globally (Singapore) and 4 months behind the first adopters in Europe (Italy and France). It is built on a decentralised system, with potential exposure to the virus being determined locally on the users' phones, minimising data sharing (see also 'Third party data access' under section 2.3 for additional details). In practice, this also means that, as opposed to direct intervention by NHS Test and Trace (human contact tracing or receipt of a positive test result), there is 'no legal duty' to self-isolate if instructed by the app (see https://faq.covid19.nhs.uk/article/KA-01398/ and https://www.nhs.uk/conditions/coronavirus-covid-19/self-isolation-and-treatment/if-youre-told-to-self-isolate-by-nhs-test-and-trace-or-the-covid-19-app/). The advice to self-isolate and count-down lasts for ten days after predicted exposure. Additional features of the app include routine venue check-ins (pubs, restaurants), local public health advice, a symptom checker—encouraging continued use.

## METHODS
### Subjects, setting and data linkage
Our panel vendor is ORB International. We use a sample of 2500 respondents across three waves of data collection. We consulted the vendor and planned attrition so that the first wave of surveys were completed by 5000 respondents in July 2020, the second wave by 3700 in November 2020, and the final wave by 2500 in March 2021—consisting of those who completed all three surveys. Our study of adoption is embedded in a larger population survey about people's lived experience in 'Brexit-Covid-19 Britain (For more information, see https://brexit-studies.org/covid-19)' thus our sample size is not determined by power calculations for this specific study. Other than compliance with quota sampling demographics (managed by panel vendor) and participation in all three waves, there were no exclusion criteria for this study. While the NHS COVID-19 app is used by citizens living in England and Wales, we needed to restrict our study to England's population on the funder's request (We used the Revised Standards for Quality Improvement Reporting Excellence (SQUIRE) reporting guidelines[12]).

We provide an overview of the study design in figure 1 below. Matching the survey dates and respondents' self-reported postal districts (first part of postcodes), we merged COVID-19 cases data, regional closure and restrictions data (three-tier system overlapping with wave 2), and urban/rural neighbourhood data from external sources, as detailed below. Our data linkages are probabilistic as neither units of analyses across the official data constitute an exact match to postal districts, however, asking for more granular location data (eg, postal area) from our respondents would have potentially compromised privacy.

Coronavirus cases are published on the UK Government's official website, and updated on a weekly basis (week's end) on the Middle Layer Super Output Area (MSOA) level. While sometimes MSOAs are entirely contained within a district constituting an exact match, often a number of these overlap with a district. For simplicity, we link data from the largest overlapping

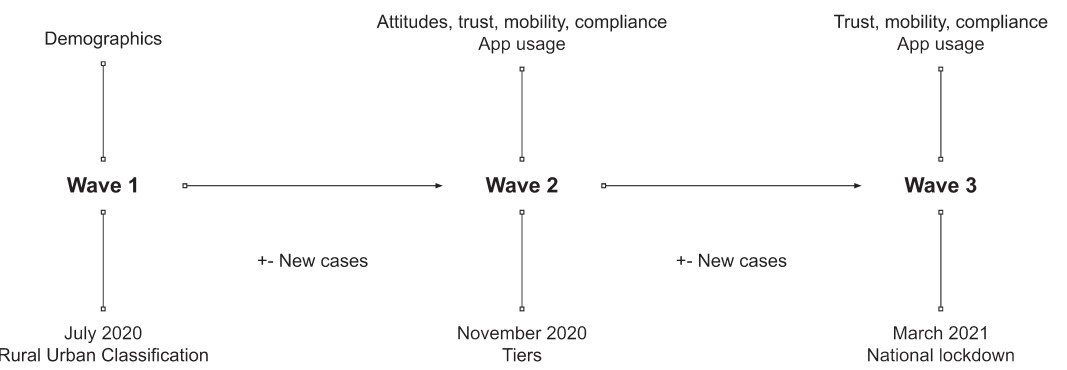

**Figure 1** Research design.

MSOA in terms of population wherever there is ambiguity (For example, people living in an EX4 postcode may be counted across thirteen MSOAs but the percentage overlap ranges between one per cent (Mid Devon 10) to 100% of MSOA located within EX4 (Exeter 002) thus we merged it with the latter. We tried an alternative method of estimating case numbers by weighting the MSOA totals according to the proportion of overlap with postcode districts and this produced similar results.).

Tiers data were published on the UK Government's official website when changes occurred, with restrictions applied on the local authority district (LAD) level. Postal districts were linked to LADs and a tier assigned where a postcode was situated wholly within a single tier. Where a postcode overlapped with LADs assigned to different tiers, we used the respondent's self-reporting of their tier.

Official urban–rural classification data (Census 2011) is available on the more granular Output Area level that are linked to postcodes by the ONS. We aggregated these into districts comprising a number of urban and rural locations, of which we took the modal category for merging with our dataset.

The linked, final dataset is publicly available, see also data availability statement.[1]

### Patient and public involvement
Patients or the public were not involved in the design, conduct, or reporting plans of this research. However, a range of public impact activities connected to the broader project about inequalities linked to COVID-19 in Britain will include a summary of this research.

### Outcomes
The dependent variable is adoption measured first in wave 2, shortly after the app's roll-out on 24 September 2020 by the Department of Health and Social Care; and again, in wave 3. We provided the following description along with a close-ended question:

Contact tracing is a tried and tested method used to slow down the spread of infectious diseases. Contact tracing can be done by public health officials or digitally with mobile phone applications or wearable devices. On the 24th of September, the government launched an NHS contact tracing app for England and Wales that will notify you if you have been in close contact with someone who has tested positive for Covid-19. Are you using this app?

Based on the responses submitted in wave 2, we split the sample for analysis in wave 3 to examine continued use separately from new adoption, see section 2.6.

### Predictors
For exact question wording and additional information about these variables, see data availability statement.

### Demographics: wave 1
Among standard sociodemographic questions were age, gender, education level, and identification with a list of 14 ethnic minority groups (including 'mixed') in addition to 'Whites', following the recommendation by the Office of National Statistics England-specific list (https://www. ons.gov.uk/methodology/classificationsandstandards/ measuringequality/ethnicgroupnationalidentityandre ligion). Location is provided as the first section of the postcode (first three to four digits), which we refer to as the postal district.

### App attitudes: wave 2
The first set of attitudinal questions, measured on five-point agreement scales about the app itself tap into four aspects of an extended version of the Technology Acceptance Model to health tech including apps and wearable devices[14] (These variables are subject to missingness, see 2.4 Analytical framework and online supplemental appendix figure A1):
► The app's perceived ease of use (or judgement whether it would be easy to use if respondent has not adopted the app yet).
► Its perceived usefulness to slow the spread of the virus.
► Whether and how concerned respondent is about privacy when using the app.
► Social norms in terms of whether people in respondents' social networks think it is a good idea to use the app.

The second set of questions expand on the above with questions adopted from the results of the qualitative work

of app users[8] on consultation with the study's authors. The study mode was a 'rapid online discussion' event with a deliberative format (deliberative poll) where 28 members of the public were selected to discuss and consider a variety of viewpoints about the app while crystallising their own opinions. The additional questions in our survey reflect the concerns that emerged from this event (p. 4)[8] and are similarly measured on five-point scales:

► Whether respondent needs transparent evidence that the app is indeed effective.
► Whether respondent needs further information about how the app treats and uses data.
► Whether respondent needs further information about how the needs of vulnerable groups (eg, older age) are addressed.

### Third-party data access: wave 2

We asked both users and nonusers 'Who do you think will have access to the data collected by the NHS COVID-19 app?' In response, they could use a checklist of up to eight items or 'none of these.' The parties were listed as follows: the NHS, UK Government, Local health authority, UK Police, Apple, Google, Your telecommunications provider, Your internet network provider. For simplicity, we use concern about privacy (see above) in our complex multilevel models predicting app usage and scrutinise privacy further in section 3.3 (see the Results section) using this measure separately. The app's primary purpose is the automation of contact tracing locally on phone, third-party access is kept to the minimum by sharing anonymised data only. Apps match a list of 'broadcast codes' and venues encountered by the app with a list curated by public health officials showing evidence of infection.[15]

### Trust: all waves

In each survey wave, we asked about general trust in government on a 0–10 scale; we predict wave 2 adoption with wave 2 trust and wave 3 continued use with wave 3 trust. We expect that the government's ability to gain and maintain its citizens' trust will motivate uptake of contact tracing apps.[3 4]

### Mobility: all waves

As for behavioural predictors, we asked a set of questions about stay at home orders including 'working from home.' This predictor draws on an influential contact study showing that high infection rates particularly in disadvantaged neighbourhoods were explained by mobility patterns due to these residents' inability to work from home.[16] The response options were 'followed 100%', 'mostly complied', 'mostly not complied', 'was not possible to comply' and 'does not apply to me'. We used working from home as a proxy of more substantial and regular mobility (for work rather than recreational purposes). We dichotomised this measure so that we obtained a group of respondents who were likely not

mobile (followed 100% or mostly complied with stay at home) and those who likely remained mobile (those who did not/could not comply in addition to those who did not need to comply).

### Compliance: all waves

Across a set of 20 questions, we asked about the ways in which respondents have been affected by the coronavirus. One of these options was 'Have worn a face mask when out in public,' which we use as a proxy for compliance with other non-pharmaceutical public health interventions to control the spread of COVID-19.

### Second-level (postal district) variables

The procedure of data linkage is described in section 2.3. District-level characteristics such as case numbers, stricter local lockdowns or higher population density in urban and metropolitan locations may affect overall anxiety and uncertainty that can generate more compliance with health interventions. We include the following measures varying across respondents' postal districts:

► The number of new cases recorded by the end of the week while the survey was in the field, available for all waves.
► The temporary local restriction tiers at the time of wave 2 coded tier 1 (medium alert), tier 2 (high alert), tier 3 (very high alert).
► The dichotomous urban location measure derived from ONS Rural-Urban Classifications data (in which a location is classified urban if 74% or more of the resident population living in urban areas).

### Analytical approach

We combine the measures listed above in two sets of multilevel logistic regression models, first estimating adoption at wave 2, then depending on wave 2 response either continued use in wave 3 or new adoption in wave 3. In all cases, we first fit null models estimating variance of uptake across second-level units (postal districts) as random intercepts, and continue to add individual and district-level predictors as appropriate. We scale and mean-centre all continuous predictors. As we observed a pattern of non-random missingness on the attitudinal predictors, we carried out multiple imputation and pooled the estimates across five imputed datasets, see online supplemental appendix for further details (As we show in online supplemental appendix figure A1), missing data particularly on ease of use is related to non-adoption thus exclusion of these cases would be inappropriate. In practice, omitting these observations had little impact on our initial uptake and new uptake models, but had an effect on the significance of three technology acceptance variables in the continued use model.). For parsimony, we analyse respondents' perceptions of third-party data access by adoption in section 3.3 separately as their inclusion in the regression models would add eight additional categorical predictors relating to a similar underlying concept (privacy).

## RESULTS

### Uptake and geographical variation

We observe uptake in 2020 November at 41%, 95% CI (0.39% to 0.43%). Of the initial adopters, 124 or 12% of respondents no longer said they used the app by wave 3, 95% CI (0.10% to 0.14%); while of those initially not adopting, 98 respondents or 7% reported usage by wave 3, 95% CI (0.05% to 0.08%) (including five who responded with 'Don't know' in wave 2). Of those not using the app in wave 2, 36%, 95% CI (0.34% to 0.38%) reported that they did not own a suitable device (This might seem high, but given (1) that iPhone 6 and earlier as well as Android 6 (Marshmallow) and earlier phones cannot run the app (https://faq.covid19.nhs.uk/article/KA-01116/en-us); (2) the high level of misinformation we report about third party data access among nonusers and (3) that notable segments of the UK population perceive the Government's COVID-19 communication as 'low' in clarity according to recent reports,[17] we think it 36% of nonusers may very well have concerns about device compatibility), 1% (16 people) that they were discouraged to use it by their employer (eg, reports in September 2020 confirmed that police officers were asked not to instal app on work phones or ignore advice on personal phones, see https://www.bbc.co.uk/news/technology-54328644), while the rest may be linked with other reasons including what we report in Section 3.2. Although not part of our theoretical framework, we note that respondents who had the coronavirus were just as likely to be using the app as not using it: 2.39% (95% CI 1.61% to 3.53%) and 2.71% (95% CI 1.99% to 3.69%), respectively. By wave 3, people who had the coronavirus were users of the app in only slightly higher proportion than non-users, 3.41% (95% CI 2.44% to 4.75%) and 2.82% (95% CI 2.10% to 3.80%), respectively, with 95% CIs.

Looking at the initial measure of adoption in wave 2, our random intercept model detects some variation across postcode districts, SD=0.34, shown in figure 2. This is similar to the magnitude of mobility and compliance effects and about half of the magnitude of the most influential attitudinal effects. By contrast, we find no variation across districts in wave 3 either for continued use or for new adoption. We explain this by comparing the two time points in terms of mobility and social contact: while in November 2020, these districts belonged to different tiers of restrictions (less open districts in tiers 2 and 3 with lower mobility thus lower adoption), in March 2021 all districts faced similar restrictions under a national lockdown. Beyond local tiers of restrictions, we find little evidence that COVID-19 case numbers influenced adoption but we found that initial enthusiasm to adopt the app was higher in urban locations.

### Individual-level predictors

We provide an overview of the results visually in figure 3 below, and summarise all fitted models and list ORs in online supplemental appendix table A1. The individual-level predictors of initial adoption in wave 2 are drawn from a multilevel model accounting for the postal district-level variation as shown above. The individual-level predictors of wave 3 continued use (subset of respondents who were adopters in wave 2) and of new adoption (subset of respondents who were non-adopters in wave 2) are drawn from simpler linear models as we found no comparable variation on the postal district-level (In both cases, the null model consisting of an intercept and random effects only, the variance component is either zero (singular) or would not reach convergence.) and thus multilevel modelling was not appropriate.

Attitudinal predictors, particularly technology acceptance model variables, appear to be the most powerful predictors of adoption. When it comes to demographics, we find older respondents less likely to be adopters in both waves but not more likely to drop out of usage.

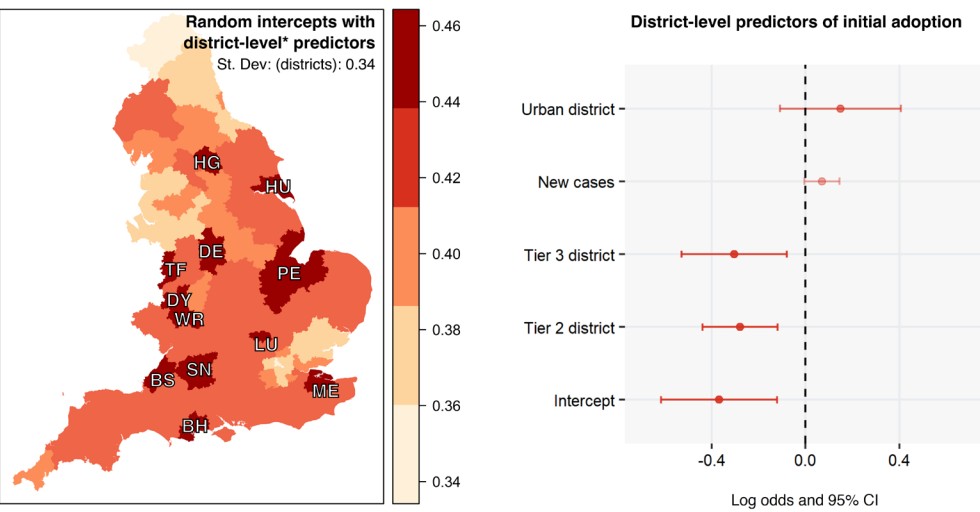

**Figure 2** Geographical variation of uptake percentage (left, plotted here across postal area eg, BH) and log odds and 95% CI of group-level predictors.

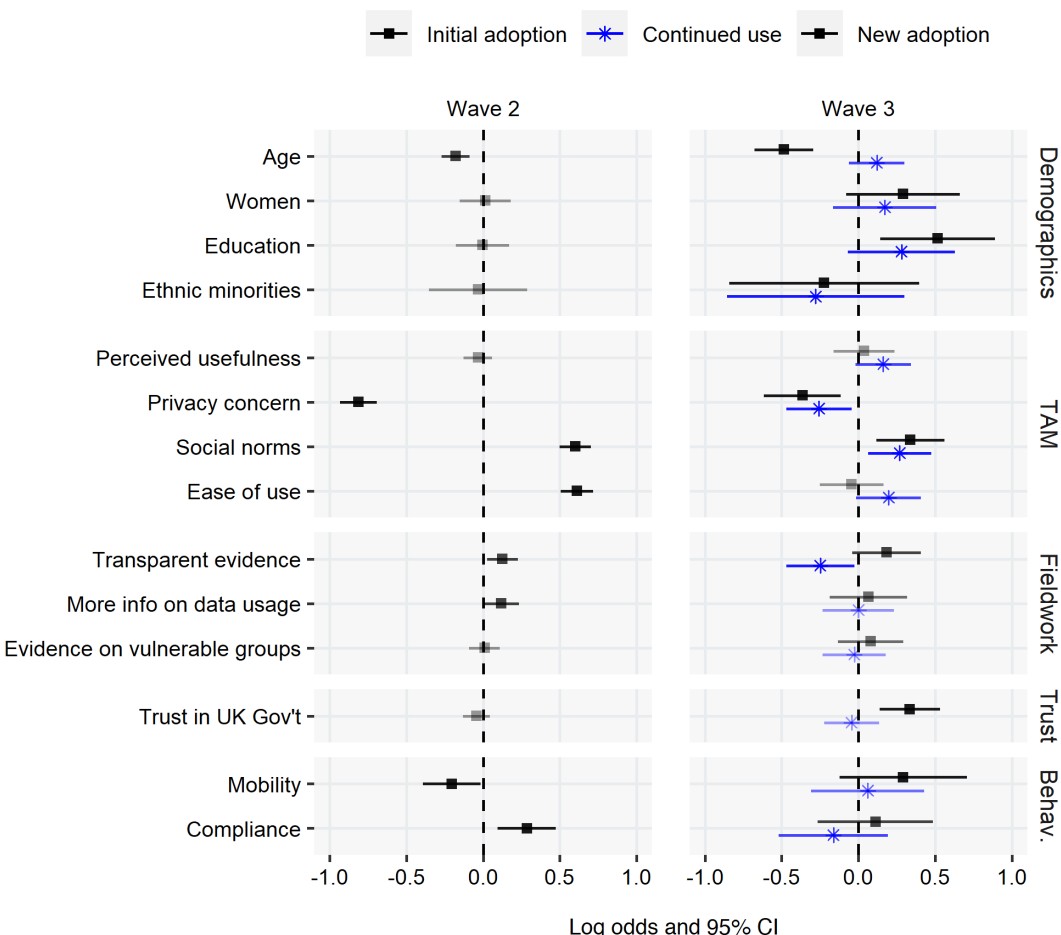

**Figure 3**  Individual-level predictors: log odds and 95% CI. TAM, technology acceptance model.

Education has an impact on new adoption in wave 3 only with respondents higher than the median education level more likely to opt into usage. We find that ethnic minority respondents were somewhat less likely to be adopters and more likely to drop out of initial usage, as opposed to those identifying with the group 'Whites' only. The small group size (8.20% of the sample) is, however, reflected in the large uncertainty around the estimate not meeting conventional thresholds of statistical significance.

In terms of technology acceptance, we find that the perceived usefulness of digital contact tracing to slow the spread of the virus is not influential on initial adoption or wave 3 new adoption. When it comes to continued use, however, respondents who thought the app was useful were somewhat more likely to continue using it by wave 3. Related variables also have large effects: people concerned about privacy were less likely to adopt the app, while those who agreed that people in their social circles (family, friends, work) thought it was a good idea to use the app were also likely to adopt. See also section 3.2 for additional insights on privacy. Perceived or expected ease of use also appears important although the direction of causality is less clear; users exposed to the app may have become more confident in its usability.

The items adopted from the deliberative poll have only small impacts on initial adoption. We expected that concern about the lack of transparent evidence would predict non-usage from the outset, but our results suggest that initial adopters of the app are more likely to think in these terms. When it comes to continued use, however, we find that those who had concern about the lack of transparent evidence were indeed more likely to drop out of usage. We find small effects regarding information on data usage with initial adopters who were more likely to miss this kind of information. We find negligible effects relating to the vulnerable groups steer. However, we find that concern about the needs of vulnerable groups is related to age and work status with retired respondents a little more likely to express concern, $t(1316.4)=-3.18$, $p<0.01$, mean difference of 0.20; and also people in the highest age group, $t(1191.9)=-2.68$, $p<0.01$, mean difference of 0.13 on a five point scale.

Of the rest of the individual-level characteristics, we note that trust in the UK government, although not influential in the first decision to adopt was predictive of new adoption in wave 3. In terms of mobility, people not working from home were less likely to adopt in wave 2 which may be cause for concern, potentially making the

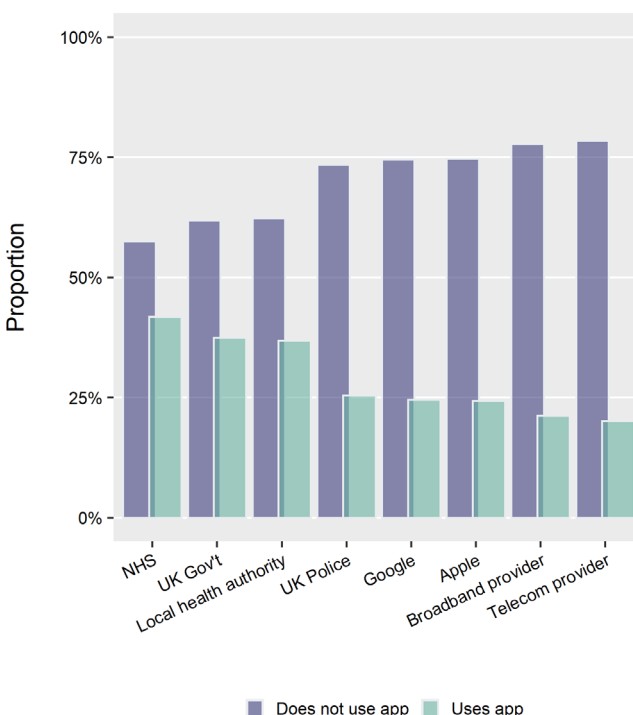

**Figure 4** Respondents' perceptions of third-party data access, wave 2. NHS, National Health Services.

app less effective, but in wave 3, this effect is no longer significant. Mask wearing predicts adoption in wave 2 only, suggestive of masks becoming increasingly normative and less contentious in the population as a whole.

### Third-party data access

Further scrutinising privacy, we ask if users and non-users have different perceptions about who has access to the data collected by the app (We also asked about the kind of data collected by the app, but only from app adopters. The top answers were: over 60% of users correctly identified venue check-ins; approximately 40% incorrectly identified exact location; approximately 20% correctly identified user-provided health data and the same percentage incorrectly identified 'contacts from phone.'). While above we found that overall 'concern about privacy' is a powerful predictor of non-adoption, figure 4 below looking at third-party data demonstrates how this concern may translate into perceptions about data access.

While a plurality of both users and non-users think the NHS has access to data collected by the app, stark differences emerge when it comes to other parties such as the police or broadband providers producing a ¼–¾ split between users and non-users. While in itself the relationship may not be causal, it is consistent with the model-based results above regarding privacy concern.

### CONCLUSION AND DISCUSSION

Our result regarding general uptake, 41% in November 2020, is consistent with other reports that followed our

data collection (see https://www.adalovelaceinstitute. org/project/covid-19-digital-contact-tracing-tracker), strengthening our survey's external validity. The observed drop-out rate between waves 2 and 3 was 12%. We note that at the timing of our follow-up survey, England was under national lockdown, thus mobility and social contact decreased overall, making the app less needed for regular use (eg, venue check-ins). Similarly, while there was a lot more overall mobility in wave 2 during initial uptake, the linked contextual variables showed that the app was less used where there were some regional restrictions. Notably, we found that new adoption (people not using the app in wave 2 but opting into usage in wave 3 of the survey, 7%) was facilitated by high trust in the UK government, in line with emerging literature linking trust to compliance during times of crisis.[18] We speculate this is linked to the increased severity of crisis including new variants by wave 3, and perceptions about the UK government's early success in its vaccination programme.

We found that attitudinal characteristics, notably social norms and privacy concerns, were powerful predictors of (non-)adoption. With additional analysis, we also found that non-users particularly overestimated the potential for third-party data access including non-health actors such as broadband and telecom providers. This contrasts with earlier experimental findings on intention to use in the UK[7] which polled respondents in June 2020 thus well before the app's roll-out, documenting little concern about varying privacy features and or potential data breaches. We speculate these differences are, on the one hand, due to study design: In the experimental scenario, respondents were given complete and transparent information about the app's data usage and storage settings, whereas in the field this information is likely to be more opaque, with citizens more risk-averse. Indeed, in this study we found transparency of evidence about effectiveness was a concern that predicted drop-out. On the other hand, the early versions of the app would have used an NHS-centralised system as opposed to the final decentralised Exposure Notifications System created by Google and Apple jointly. While the latter has better overall privacy preserving characteristics, high public trust in the NHS may have mitigated concern about privacy more effectively.

Among other variables, we found work-related mobility is also associated with adoption[16] which partially explains why indicators of structural inequality (eg, ethnicity) are less relevant. Yet, after controlling for mobility, ethnic minority participants are still slightly less likely to adopt, a phenomenon that might be better explained with targeted data collection with a larger pool of ethnic minority respondents.

Our findings contribute to the understanding of inequalities around the adoption and public acceptance of digital technologies supporting the public health response to the pandemic globally, extending beyond contract tracing to telemedicine, digital health passports or targeted public health messaging. To tap inequalities,

we aimed at an explanatory model with an exhaustive set of demographic, attitudinal, behavioural and postal district-level characteristics.

We acknowledge that our study is limited to a sample of English residents and thus not fully representative of app usage in both England and Wales, where the NHS COVID-19 App was released; nor of health behaviour more generally across the UK. In England, overall mobility was restricted during wave 3 of our survey which allowed limited opportunities for app usage such as venue check-ins (venues were closed). In addition, data collection in the future would benefit from a booster sample of ethnic minority participants to better understand inequalities across diverse ethnic populations: Our results suggest there may be some ethnicity effects on adoption even after controlling for a range of other predictors, however, we lack sufficient numbers to investigate these mechanisms.

**Acknowledgements** We thank Cary Kind and The Ada Lovelace Institute for recommending questions to include from their deliberative poll.

**Contributors** LH is the author responsible for the overall content as the guarantor. All authors contributed to the study concept and design, and the development of the questionnaire. LH, SB, AJ and DS have full access to the data and have performed analysis leading to the results reported in this paper. LH, SB, JB, CD, AJ, OJ, DS and KT contributed to the interpretation of results, writing and critical revisions. All authors approved the final version to be published and are accountable for all aspects of the work.

**Funding** This research is funded by the Economic and Social Research Council as part of UK Research and Innovation's rapid response initiative to COVID-19. The project is entitled: 'Identity, Inequality and the Media in Brexit-Covid-19-Britain' (Grant Ref: ES/V006320/1). Project website: https://www.brexit-studies.org/covid-19.

**Map disclaimer** The inclusion of any map (including the depiction of any boundaries therein), or of any geographic or locational reference, does not imply the expression of any opinion whatsoever on the part of BMJ concerning the legal status of any country, territory, jurisdiction or area or of its authorities. Any such expression remains solely that of the relevant source and is not endorsed by BMJ. Maps are provided without any warranty of any kind, either express or implied.

**Competing interests** None declared.

**Patient consent for publication** Not applicable.

**Ethics approval** This study was approved by Prior to data collection, our research design received ethical approval from the University of Exeter College of Social Sciences and International Studies Ethics Committee on 16 July 2020, Certificate No. 201920-131. Our panel vendor ORB International manages informed consent from their panel respondents.

**Provenance and peer review** Not commissioned; externally peer reviewed.

**Data availability statement** Data are available in a public, open access repository. The questionnaires, case summaries, as well as replication data are available on figshare: Stevens D, Banducci S, Horvath L, Jones A. Identity, Inequality, and the Media in Brexit-COVID-19-Britain Surveys, Waves 1–3. figshare; 2021 (cited 9 May 2021). https://doi.org/10.6084/m9.figshare.14527188.V.1

**ORCID iD**
Laszlo Horvath http://orcid.org/0000-0003-0606-1050

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
