## [Reviewer comments · BMJ Open]

ARTICLE DETAILS

TITLE (PROVISIONAL)	Adoption and continued use of mobile contact tracing technology: Multilevel explanations from a three-wave panel survey and linked data
AUTHORS	Horvath, Laszlo; Banducci, Susan; Blamire, Joshua; Degnen, Cathrine; James, Oliver; Jones, Andrew; Stevens, Daniel; Tyler, Katharine

VERSION 1 – REVIEW

REVIEWER	Katherine Blondon University Hospitals of Geneva Department of Internal Medicine, Division of General Internal Medicine
REVIEW RETURNED	29-Aug-2021

GENERAL COMMENTS	This paper is generally well written, explicit and rigorous in terms of methods, but does not seem to address the question fully. The reader lacks information about how the studied contact tracing app in the UK works. For example, contact tracing apps in some countries was inactivated once the user tested positive. Was this the case in the UK? Or was it kept active to help trace the new variants? These points are unclear in the analysis. Furthermore, how did getting covid affect the users' app use? The authors report 36% of non-users due to lack of appropriate device. Doesn't this seem excessive? Contact tracing apps (again, I am assuming the UK version was like others) were generally designed to be compatible with the majority of smartphone devices... The organisation of the text can be improved: the introduction should provide an overview of the paper, not go into detail about the results of the study (e.g. last paragraph of page 2). The conclusion of the paper, quite clear in the abstract is a little confused at the end of the paper, and would be much improved with a final conclusion after the discussion paragraph. Finally, a question about methods, and future perspectives: it would have been more interesting to have access to log data of app use, rather than two self-reports of app use. Is there any possibility of this in future research?
---

REVIEWER	Eskinder Eshetu Ali Addis Ababa University College of Health Sciences
REVIEW RETURNED	13-Sep-2021

GENERAL COMMENTS	Comments on: Adoption and continued use of mobile contact tracing technology: Multilevel explanations from a three-wave panel survey and linked data
--

	Journal: BMJ Open Overall comment: I think the topic is relevant and the sample size involved is good to make generalisations. However, I have some specific comments that the authors need to address before publication. Specific comments Abstract:  • The design/setting section does not tell the reader about what stats tests were used to establish the relationship between adoption/continued use and the independent variables. • The results section seems to be only descriptive. Only percentages and CI are reported. Given the sample size of the data and repeated measures, Authors need to show relevant statistic with p-values. • It would be better to make the conclusion succinct [finish in one or two sentences]. That way the word count saved could be used to make the methods and results sections better. Strengths and limitations:  • The second bullet point lacks clarity. Looks like it is strength, but is unclear. Please make it to the point. Introduction:  • The last sentence of the second paragraphs mentions “section 2.5” but there is no such section in the manuscript. The same assertion was also made under section 2.4. Please revisit • Some details are better to be put under the methods section instead of the introduction section. This included details such as: how data were collected, time of data collection... etc (much of paragraph three) • Some details are better to be put under the results section instead of the introduction section. E.g. paragraphs four and five... most, if not all, of the details are those actually put under the results section of the abstract. This makes your introduction a little confusing and is sort of jumping the gun to talk about issues that should have been in the results. • I did not see much about the NHS app, which would have been interesting to read in the introduction. Methods:  • It is not clear how the sample size was determined. Why 2500, 3700 or 5000? Was there a statistical sample size calculation formula employed? Is it just a guess? Or what could be collected given the available recourses for the authors? Or is it all the data available? • The authors were not specific enough about the sample used in the analysis. Is the 2500 participants those who persisted throughout the three time periods of data collection? It appears so in 2.1 and fig 1, but not clearly specified. This needs to be clarified. • I don't understand why the authors resorted to footnotes to clarify some issues. I think all clarifications should be within the main text and there is no need to use footnotes (not just in the methods section but also throughout the manuscript except the case of figures and tables) • The methods section does not say much about the statistical test techniques that were used in the analysis. The authors should describe:  o What statistical tests were used
--	--

	 o What diagnostic techniques were followed to evaluate whether the data were in line with the assumptions of the selected statistical test o What was the level of significance chosen o Other information relevant to the selected test. • I have not found any information about ethical aspects of the study. Did the authors receive ethical approval? If ethical approval was not necessary, did they get a waiver of ethical approval? Such things should be clearly described under the methods section. Results:  • There was no reference to Figure 2 under paragraph 1 section 3.1. • The major issue about the results is that it is not clear what the stats test is and is difficult for the reader to understand the meaning of each of the findings. The figures are not clear enough as a result.
--	---

VERSION 1 – AUTHOR RESPONSE

Reviewer: 1

Dr. Katherine Blondon, University Hospitals of Geneva Department of Internal Medicine

This paper is generally well written, explicit and rigorous in terms of methods, but does not seem to address the question fully. The reader lacks information about how the studied contact tracing app in the UK works. For example, contact tracing apps in some countries was inactivated once the user tested positive. Was this the case in the UK? Or was it kept active to help trace the new variants? These points are unclear in the analysis.

We thank the Reviewer for the very detailed and helpful feedback, and are happy to take on all recommendations. We now provide a more general description of the UK app now quite early on in Section 1.1. This complements a brief discussion of third-party data access in Section 2.3. Re active/inactive apps: we note in this new section that, apart from predicting exposure to the virus, the app advises on self-isolation, features "routine venue check-ins (pubs, restaurants), local public health advice, a symptom checker—encouraging continued use." (from last sentence of Section 1.1)

Furthermore, how did getting covid affect the users' app use?

In wave two, respondents who had coronavirus were just as likely to be using the app as not using it, 2.39% [1.61,3.53] and 2.71% [1.99,3.69] respectively, with 95% CIs. By wave three, people who personally had coronavirus were users of the app in only slightly higher proportion than non-users, 3.41% [2.44,4.75] and 2.82% [2.10,3.80], respectively, with 95% CIs. We are now reporting these statistics, as stated here, in the first paragraph of Section 3.1 Uptake and geographical variation.

The authors report 36% of non-users due to lack of appropriate device. Doesn't this seem excessive? Contact tracing apps (again, I am assuming the UK version was like others) were generally designed to be compatible with the majority of smartphone devices...

To reflect on this, we added the following where the results are reported in Section 3.1: Our result regarding general uptake is consistent with other reports e.g. <https://www.adalovelaceinstitute.org/project/covid-19-digital-contact-tracing-tracker/> but perception, as we ask, about compatibility of one's device is difficult to benchmark to other

sources. However, given (1) that iPhone 6 and earlier as well as Android 6 (Marshmallow) and earlier phones cannot run the app (<https://faq.covid19.nhs.uk/article/KA-01116/en-us>); (2) the high level of misinformation we report about third party data access amongst nonusers; and (3) that notable segments of the UK population perceive the Government's Covid-19 communication as "low" in clarity according to recent reports (38%, Abrams et al. 2021), we think it 36% of nonusers may very well have concerns about device compatibility.

The organisation of the text can be improved: the introduction should provide an overview of the paper, not go into detail about the results of the study (e.g. last paragraph of page 2).

Thank you for the suggestion, we significantly cut down the last two paragraphs (from 301 words to 131 words) of the Introduction.

The conclusion of the paper, quite clear in the abstract is a little confused at the end of the paper, and would be much improved with a final conclusion after the discussion paragraph.

We agreed, and corrected this: structured it better across multiple paragraphs instead of the two before and followed the order in which these results were presented in the abstract. Re conclusion: we now first state a concluding sentence summarising the main findings, and then proceed to discuss the implications. We moved the contributions of the study to the end. The section is now called Conclusion and Discussion.

Finally, a question about methods, and future perspectives: it would have been more interesting to have access to log data of app use, rather than two self-reports of app use. Is there any possibility of this in future research?

This is an interesting suggestion and we think tracking data can give a more precise log of app usage, however, we would still use that in combination with survey data such as ours, to be able to tap attitudinal predictors of uptake and continued use (e.g. trust in government) which is more useful for policy recommendations. Tracking log data alone does not give us an insight into people's perceptions of app usage.

Reviewer: 2

Dr. Eskinder Eshetu Ali, Addis Ababa University College of Health Sciences

I think the topic is relevant and the sample size involved is good to make generalisations. However, I have some specific comments that the authors need to address before publication.

We thank the Reviewer for these comments which are much appreciated. We hope we managed to address each of them carefully.

Specific comments

Abstract:

- The design/setting section does not tell the reader about what stats tests were used to establish the relationship between adoption/continued use and the independent variables. We added this information to the abstract. We keep the 'design' heading to summarise research design (survey waves) only. Instead, we added a new heading 'Analytical approach,' mirroring Section 2.4 where the details are discussed. In turn, in Section 2.4 we again clarified that these are multilevel logistic models---indeed before we only referred to its special cases: null models with random intercepts, then extension with individual and district-level predictors. See Gelman and Hill 2006.
- The results section seems to be only descriptive. Only percentages and CI are reported. Given the sample size of the data and repeated measures, Authors need to show relevant statistic with p-values. We hope that our response to the previous comment - and solution provided - will convince the Reviewer that the results are not descriptive, nor presented as such. **In subsection 3.1** we show information about the dependent variable, which is indeed percentages but also 95% confidence intervals (thus not descriptive and states the level of significance) as preferred by BMJ Open - but we realise we needed to add CI to a few more points that we missed. **Then in subsection 3.2**, we discuss the effect sizes from our multilevel models. It is not normally expected that we discuss each logistic regression coefficient one by one in the main text - there are 19 per model. Instead, as conventional, we show these visually in the figures, and report them as log odds, odds ratios, along with standard errors and p values in tables. See Table A1. We point this out in our first sentence in Section 3.2: "We provide an overview of the results visually in Figure 3 below, and summarise all fitted models and list odds ratios in Table A1 in the Appendix." **In Figures 2 and 3**, we updated the plot labels to imply that we kept the 95% CIs in these too.
- It would be better to make the conclusion succinct [finish in one or two sentences]. That way the word count saved could be used to make the methods and results sections better. We were asked by the other reviewer to fix the conclusion section as well and agree that this needed to be addressed. The conclusion now follows the following format: one takeaway per paragraph, in the first sentence, making them more succinct. The contributions of the article are now moved to the end of the section. We hope this will address both Reviewers' concerns at the same time.

Strengths and limitations:

- The second bullet point lacks clarity. Looks like it is strength, but is unclear. Please make it to the point. We reformulated this - indeed a strength but clarified what the point was. Thank you!

Introduction:

- The last sentence of the second paragraphs mentions “section 2.5” but there is no such section in the manuscript. The same assertion was also made under section 2.4. Please revisit

This was a mistake that we now corrected. Thank you!

- Some details are better to be put under the methods section instead of the introduction section. This included details such as: how data were collected, time of data collection... etc (much of paragraph three)

We agree that these details can be cut down. Our intention was to give an overview in the Introduction, and some of the study design will be part of that (representative survey, timing of waves we believe are still important) but we deleted some of the details about the variables - we hope this addresses the concern raised in this point.

- Some details are better to be put under the results section instead of the introduction section. E.g. paragraphs four and five... most, if not all, of the details are those actually put under the results section of the abstract. This makes your introduction a little confusing and is sort of jumping the gun to talk about issues that should have been in the results.

We corrected this, also raised by Reviewer 1, especially the last two paragraphs of the introduction are now significantly cut down, from 301 words to 131 words.

- I did not see much about the NHS app, which would have been interesting to read in the introduction.

Thank you for the suggestion, we included a separate subsection 1.1 in the introduction to add details.

Methods:

- It is not clear how the sample size was determined. Why 2500, 3700 or 5000? Was there a statistical sample size calculation formula employed? Is it just a guess? Or what could be collected given the available resources for the authors? Or is it all the data available?

We now clarified this under 2.1 Subject, setting : "Our study of adoption is embedded in a larger population survey about people's lived experience in 'Brexit-Covid-19 Britain' thus our sample size is not determined by power calculations for this specific study." Indeed, as the Reviewer suspects, this is part of the resources we requested from our funder, UK Research and Innovation.

- The authors were not specific enough about the sample used in the analysis. Is the 2500 participants those who persisted throughout the three time periods of data collection? It appears so in 2.1 and fig 1, but not clearly specified. This needs to be clarified.

Indeed, this is a sample of 2,500 respondents across three waves - to improve clarity, we worked on the phrasing in the first sentences under Methods, closing with "the final wave by 2,500 in March 2021—consisting of those who completed all three surveys."

- I don't understand why the authors resorted to footnotes to clarify some issues. I think all clarifications should be within the main text and there is no need to use footnotes (not just in the methods section but also throughout the manuscript except the case of figures and tables)
 - We did not find any indication on the thorough BMJ Open guidelines that footnotes should be avoided. We use them to discuss details that are not related to the main points, in addition to pointing to non-academic references (websites, for example).
- The methods section does not say much about the statistical test techniques that were used in the analysis. The authors should describe:
 - What statistical tests were used - addressed, see our response above
 - What diagnostic techniques were followed to evaluate whether the data were in line with the assumptions of the selected statistical test - a key check on multilevel models is the specification of the null model - is there second-level variation? We performed these, and informed our model choice, as described. In addition, we performed analyses of missingness patterns, which we report in the Appendix. As the model type is binomial, rather than linear, we do not have much in terms of additional diagnostics to perform.
 - What was the level of significance chosen - addressed, see our response above.
 - Other information relevant to the selected test - we hope our clarified description provides all relevant information.
 - I have not found any information about ethical aspects of the study. Did the authors receive ethical approval? If ethical approval was not necessary, did they get a waiver of ethical approval? Such things should be clearly described under the methods section. - We are strictly following the BMJ Open guidelines, which asks this to go separately in the Ethics statement, which we provided along with a certificate number, as well as a data availability statement.

Results:

- There was no reference to Figure 2 under paragraph 1 section 3.1. - That's because it is referenced in the paragraph below it. For clarity, we moved the figure placeholder beneath that paragraph.
- The major issue about the results is that it is not clear what the stats test is and is difficult for the reader to understand the meaning of each of the findings. The figures are not clear enough as a result. - Again, we hope these were addressed in our earlier comments.

VERSION 2 – REVIEW

REVIEWER	Katherine Blondon University Hospitals of Geneva Department of Internal Medicine, Division of General Internal Medicine
REVIEW RETURNED	28-Nov-2021

GENERAL COMMENTS	I thank the authors for addressing my prior concerns. I am satisfied with the revisions proposed. In my current review of the paper, I would like to raise one final point about study limitations, as I see no specific discussion about this in the paper. All studies have limitations, and besides the initial paragraph about limitations and strengths, I think this point merits some discussion, and surely has an impact on the conclusions one can draw.
--

VERSION 2 – AUTHOR RESPONSE

Reviewer: 1

Dr. Katherine Blondon, University Hospitals of Geneva Department of Internal Medicine

- I thank the authors for addressing my prior concerns. I am satisfied with the revisions proposed. In my current review of the paper, I would like to raise one final point about study limitations, as I see no specific discussion about this in the paper. All studies have limitations, and besides the initial paragraph about limitations and strengths, I think this point merits some discussion, and surely has an impact on the conclusions one can draw.

We are happy to learn that the Reviewer found our revisions satisfactory. Regarding the point about study limitations particularly sampling, we added the following paragraph on p. 10 of the marked copy: *We acknowledge that our study is limited to a sample of English residents and thus not fully representative of app usage in both England and Wales, where the NHS Covid-19 App was released; nor of health behaviour more generally across the UK. In England, overall mobility was restricted during wave three of our survey which allowed limited opportunities for app usage such as venue check-ins (venues were closed). In addition, data collection in the future would benefit from a booster sample of ethnic minority participants to better understand inequalities across diverse ethnic populations: Our results suggest there may be some ethnicity effects on adoption even after controlling for a range of other predictors, however, we lack sufficient numbers to investigate these mechanisms.*